# Current Understanding of Human Polymorphism in Selenoprotein Genes: A Review of Its Significance as a Risk Biomarker

**DOI:** 10.3390/ijms25031402

**Published:** 2024-01-24

**Authors:** Roberto Rodrigues Ferreira, Regina Vieira Carvalho, Laura Lacerda Coelho, Beatriz Matheus de Souza Gonzaga, Maria da Gloria Bonecini-Almeida, Luciana Ribeiro Garzoni, Tania C. Araujo-Jorge

**Affiliations:** 1Laboratory of Innovations in Therapies, Education and Bioproducts, Oswaldo Cruz Institute (LITEB-IOC/Fiocruz), Oswaldo Cruz Foundation (Fiocruz), Avenida Brasil 4365, Manguinhos, Pav. Cardoso Fontes, Sala 64, Rio de Janeiro 21040-360, Brazil; reginacarvalho_rvc@yahoo.com.br (R.V.C.); llacerdac@gmail.com (L.L.C.); biagonzaga04@hotmail.com (B.M.d.S.G.); luciana.garzoni@ioc.fiocruz.br (L.R.G.); 2Laboratory of Immunology and Immunogenetics, Evandro Chagas National Institute of Infectious Diseases, Oswaldo Cruz Foundation, Avenida Brasil 4365, Manguinhos, Rio de Janeiro 21040-360, Brazil; gloria.bonecini@ini.fiocruz.br

**Keywords:** selenium, selenoprotein, biomarker, polymorphism

## Abstract

Selenium has been proven to influence several biological functions, showing to be an essential micronutrient. The functional studies demonstrated the benefits of a balanced selenium diet and how its deficiency is associated with diverse diseases, especially cancer and viral diseases. Selenium is an antioxidant, protecting the cells from damage, enhancing the immune system response, preventing cardiovascular diseases, and decreasing inflammation. Selenium can be found in its inorganic and organic forms, and its main form in the cells is the selenocysteine incorporated into selenoproteins. Twenty-five selenoproteins are currently known in the human genome: glutathione peroxidases, iodothyronine deiodinases, thioredoxin reductases, selenophosphate synthetase, and other selenoproteins. These proteins lead to the transport of selenium in the tissues, protect against oxidative damage, contribute to the stress of the endoplasmic reticulum, and control inflammation. Due to these functions, there has been growing interest in the influence of polymorphisms in selenoproteins in the last two decades. Selenoproteins’ gene polymorphisms may influence protein structure and selenium concentration in plasma and its absorption and even impact the development and progression of certain diseases. This review aims to elucidate the role of selenoproteins and understand how their gene polymorphisms can influence the balance of physiological conditions. In this polymorphism review, we focused on the PubMed database, with only articles published in English between 2003 and 2023. The keywords used were “selenoprotein” and “polymorphism”. Articles that did not approach the theme subject were excluded. Selenium and selenoproteins still have a long way to go in molecular studies, and several works demonstrated the importance of their polymorphisms as a risk biomarker for some diseases, especially cardiovascular and thyroid diseases, diabetes, and cancer.

## 1. Introduction

Selenium (Se) is an essential trace element that was first isolated in 1817 by the Swedish chemist Jacob Berzelius, who named it after a Greek moon goddess called Selene in consideration of its similarity to the element tellurium, which means Earth in Latin [1,2]. Even though it is associated with the oxygen, sulfur, and tellurium group in the periodic table, its importance was only highlighted in 1957 when Klaus Schwarz described that Se was not toxic, diverging from the widespread scientific concern at the time [3]. Schwarz’s studies demonstrated that Se is an essential nutrient that could prevent the development of hepatic necrosis caused by a diet deficient in vitamin E in rats [3,4].

Se can be found in inorganic forms as selenate and selenite and in organic forms as selenomethionine (SeMet) and selenocysteine (Sec) [5]. Once ingested in the daily diet, its inorganic form goes through the intestinal membrane, where the sodium-facilitated system absorbs it. Se is transported to the liver to participate in protein synthesis [6]. Its main organic form (Sec) is incorporated into selenoproteins, which are synthesized on their tRNA, by co-translational mechanisms for recording the stop signal UGA codon [7,8]. 

Se deficiency is a problem that affects more than 500 million people worldwide [9]. Low Se status is currently a nutritional public problem in Ethiopia and a few countries in Europe and Asia, related to an increased risk of mortality [10,11,12,13,14]. Studies clarified the influence of a poor Se diet, showing that it can affect the mRNA levels of thirteen selenoprotein genes in the spleen of pigs [15]. Se deficiency in human uterine muscle cells led to the downregulation of nineteen selenoproteins and increased intracellular ROS, cell apoptosis, and necroptosis [16]. Se deficiency causes the worsening of redox imbalance and oxidative damage to cell membranes, which could potentiate several diseases and affect the mitochondrial mechanism, inducing renal, heart, intestine, and muscle injury [17,18,19,20]. Se deficiency and immunity are associated with the frequent new strains of the influenza virus in China [9]. In addition, low levels of Se in the blood and increased levels of oxidative stress have been shown in patients with COVID-19 infection, and its severity was increased in patients who died compared to those who survived [21,22]. This alteration was described in other viral diseases, such as Ebola and HIV-1, as a risk factor for increased mortality [23]. One of the reasons for the low level of Se in the body is inadequate intake. Still, recent studies observed the impact of selenoproteins’ polymorphisms in healthy individuals and patients with specific diseases [9].

Currently, it is known that there are twenty-five selenoproteins in the human genome, such as thioredoxin reductases, selenophosphate synthetase, glutathione peroxidases, iodothyronine deiodinases, and other selenoproteins, yet some of those still do not have a well-established function [8,24,25,26] (Table 1, Figure 1).

Here, we aimed to describe the role of selenoprotein and its gene polymorphisms (Table 2). To this end, we elaborated the hypothesis: how these polymorphisms would be associated with the risk or susceptibility to developing some diseases, such as cardiovascular and thyroid diseases, diabetes, and cancer.

## 2. Glutathione Peroxidases

The glutathione peroxidases (GPXs) have an essential role against oxidative damage from lipidic peroxidases by redox reaction catalysis and hydrogen peroxide transformation, and they protect blood vessels from oxidative stress and inflammation. GPXs are critical for normal brain function, and the imbalance of GPXs could lead to impaired cognitive function and neurological disorders. For example, in Parkinson’s disease, suppressed expression of selenoprotein mRNAs was already described in the cerebellum, cortex, hippocampus, and pons. In mammals, there are five GPX selenoenzymes, called GPX1, GPX2, GPX3, GPX4, GPX6, and three other homologs where the selenocysteine site is replaced by cysteine, called GPX5, GPX7, and GPX8 [130]. Glutathione peroxidase 1 was the first identified selenoprotein in animals in 1973 [21], and it is one of the most abundant proteins of the GPX family, found in multiple subcellular locations, such as cytosol and mitochondria. The initiation transcription of GPX1 occurs by the same site in the liver, kidney, and erythroblast [131]. Studies have shown that GPX1 is more effective in detoxifying H2O2 by glutathione (GSH) oxidation than catalysis under physiological conditions [132]. GPX2 is mainly expressed in the gastrointestinal mucosa for epithelium protection against oxidative stress and to guarantee mucosal homeostasis of the gut microbiota or ingested prooxidants. GPX2 is also detectable in the liver [27]. GPX3 is widely used as a marker for Se status in plasma because it is the major extracellular isoform; its primary source is the kidney, but it is also found in breast milk and bronchoalveolar lavage fluid [133,134]. GPX3 also regulates the antioxidative effects of retinoic acid, implying that the enzyme levels may act in the viability of the human skeletal muscle stem cells [28]. This selenoprotein can act as a tumor suppressor and pro-survival protein during tumor progression in different types of cancer [29]. GPX4 works against mitochondrial oxidative damage by reducing the hydroperoxide fatty acids inhibiting lipid peroxidation in the phospholipids’ membrane [135]. The conversion of lipid hydroperoxides to lipid alcohols prevents the iron-dependent formation of toxic lipid ROS, resulting in the regulation of ferroptosis [30]. GPX4 downregulation increases the extracellular matrix degradation through the MAPK/NF-κB pathway and the sensitivity of chondrocytes to oxidative stress [136]. Its inducible disruption leads to acute renal failure and cell death in mouse embryonic fibroblasts [31]. Studies have also shown that this enzyme can play an essential role in male fertility: Se deficiency changed the protein structure of the mature spermatozoa by its ability to use hydroperoxides to form the spermatozoon [32,137]. GPX6 is in embryonic tissues and olfactory epithelium, but its function is still poorly established in humans [27,130]. In porcine, GPX6 may protect against oxidative stress in sperm capacitation by preventing the sperm from premature capacitation [33]. 

## 3. Selenoproteins 

Selenoprotein P (SelP, SELENOP) is the only selenoprotein with multiple selenocysteine residues, in contrast to others with only one or two residues [34]. This protein earned its nomenclature P from its presence in plasma [35]. SelP is synthesized in the liver, and its role is to transport Se to several tissues, maintaining cellular Se homeostasis [138]. The variation of this selenoprotein can lead to pathophysiological conditions as consequences and even as drivers, such as in type 2 diabetes mellitus, by the correlation with insulin resistance, hyperglycemia, and pulmonary arterial hypertension [36]. Deleting the SelP gene in mice revealed morphologic changes in the brain with decreased spine density and dendritic length, which could contribute to shortcomings [37]. Selenoprotein F (SelF, SELENOF) has a molecular mass close to 15 kDa in the endoplasmic reticulum [38]. The role of SelF in this process is not known. However, the protein may be involved in the unfolded protein response [139]. This selenoprotein is also associated with the plasma membrane in epithelial cells, and its levels are decreased in prostate cancers in African men as compared to normal tissue adjacent to the tumor [140]. Selenoprotein S (SelS, SepS1, SELENOS) plays a key role in the production of inflammatory cytokines, and its expression is induced by endothelial reticulum stress. The protein 50 SelS has a cytosolic tail with a coil domain that suggests it could bind other proteins for dimerization and anchor them to the endothelium reticulum membrane to maintain the protein [141]. Studies have shown that selenoprotein N is in the endothelium reticulum and is expressed in skeletal muscle, the brain, the lungs, and the placenta [142]. Selenoprotein W is localized in the mitochondria, and it is especially expressed in skeletal muscle [143]. SELENOW plays an important role during the progression of early inflammation, including arginine and tyrosine metabolism in macrophages. The loss of this selenoprotein in macrophages demonstrates involvement in metabolic reprogramming by altering the TCA cycle and glycosis [144]. Furthermore, SELENOW activates the epidermal growth factor by suppressing the epidermal growth factor receptor ubiquitination [40]. SELENOW stimulates osteoclastogenesis via activation of NF-κB and NFATc1. Its deficiency or overexpression could cause abnormalities in bone remodeling [145]. The selenoproteins W and V exhibit the same gene structure. Still, selenoprotein V is mostly expressed in the testis of rats, especially at reproductive ages, suggesting their role in male reproduction [41,146]. SELENOV has potential redox function by the impact on redox enzyme gene expression. The overexpression of this selenoprotein in 293T cells (human embryo kidney cells) induced resistance to pro-oxidant cytotoxicity. At the same time, the protein knockout in mice impacted the cell viability respiration and ATP production of the primary hepatocytes [42]. Moreover, the protein–protein interaction between SELENOV and O-GlcNAc transferase (OGT) was observed once the knockout murine demonstrated decreased activity and function of the OGT protein in fatty tissues. These data suggest a regulatory role of the O-GlcNAcylation [43]. Selenoprotein T is localized in the endoplasmic reticulum. Studies have shown that this protein could have a role in cell adhesion, and its loss elevates the expression of selenoprotein W [147]. Selenoprotein I (SelI, SELENOI) has been interpreted to play a role in motor neuron function, and it may be involved in the biosynthesis of phosphatidylethanolamine. This protein is expressed in all tissues, especially the brain, placenta, liver, and pancreas [148]. A study with a patient with severe complicated hereditary spastic paraplegia and sensorineural complications born from a consanguineous Arab Muslim family demonstrated the role of the SELENOI in the brain development, myelin formation, and preservation of ether-linked phospholipids [149]. Evidence also revealed the involvement of SELENOI in murine embryogenesis, leading to early-stage embryonic lethality in knock-out mice [150]. Selenoprotein K is in the endoplasmic reticulum and at the plasma membrane, and it is expressed especially in the heart and skeletal muscle [151]. This selenoprotein acts as a cofactor during palmitoylation by binding to endoplasmic reticulum-associated protein degradation components. In addition, SelK is important for promoting calcium flux during immune cell activation [152]. Little is known about this protein, but its overexpression suggests it has an antioxidant function [151]. Selenoprotein M may play a role in calcium regulation and protect against oxidative damage [153]. Selenoprotein R (SelR), also known as methionine sulfoxide reductase B1 (MsrB1), is a protein localized in the cytosol and in the cell nucleus that is responsible for the catalysis of methionine sulfoxide to methionine. Its active form is mainly located in the liver [154,155]. Studies suggested that it might play a role in regulating redox homeostasis and protecting the human lens epithelial cells of mitochondria [154]. In human bone osteosarcoma epithelial cells, this selenoprotein regulates proliferation by affecting the epithelial–mesenchymal transition and the mitogen-activated protein kinase pathway, corroborating previous findings [156]. Selenoprotein O (SelO) is a pseudokinase found in the mitochondria that catalyzes the addition of AMP from ATP to a protein substrate known as AMPylation [157,158]. Moreover, SelO may be essential in chondrocyte proliferation and differentiation [159].

## 4. Thioredoxin Reductases

Thioredoxin reductases (TrxR) are the only enzymes that catalyze the oxidized thioredoxins (Trx). They can directly regulate multiple gene expressions by redox signaling factors, including the protein NF-kB, apoptosis-regulating kinase, tumor suppressor gene P53, and the transcription factor AP1 [160,161]. Mammals have three isozymes: cytosolic TR1 (TrxR1), mitochondrial TR3 (TrxR2), and thioredoxin and glutathione reductase (TGR, TrxR3, TR2) [161,162]. TrxR1 and TrxR2 have broad substrate specificity, including thioredoxin, selenite, DTNB, and alloxan [163]. Both proteins are essential for the normal development of the cells during embryogenesis in most tissues. Its inactivation or deletion in mice has shown severe growth retardation and early embryonic death, which happens at the same time as the maturation of the mitochondria occurs [164,165]. The TrxRs deficiency leads to mitochondrial dysfunction, increases cell sensitivity by releasing H2O2, and exhibits DNA damage and consequent cell death [166,167]. In addition, it was demonstrated that TrxR2 deficiency could also stimulate the expression of extracellular matrix genes, increase S and G2/M phase cell distribution, leading to acceleration of the cell cycle progression, and raise chondrogenic differentiation and cartilage glycosaminoglycans [168]. Furthermore, TrxR3 deficiency alters redox status and bioenergetics during sperm maturation, capacitation, and fertilization [169]. 

## 5. Iodothyronine Deiodinases

Iodothyronine deiodinases (DIOs) activate and inactivate the thyroid hormones by reductive deiodination [170]. There are three DIOs: DIO1 and DIO3, located on the plasma membrane, and DIO2, located in the endoplasmic reticulum [171,172]. Even though DIO1 and 2 are responsible for the activation of thyroid hormone by converting T4 to T3 and DIO3 inactivates T3, DIO1 is associated with the equilibrium of plasma T3, being stimulated in hyperthyroidism and decreased in hypothyroidism, and DIO2 is associated with a ready access to the nuclear receptors for T3 production [173,174]. Deiodinases are essential for the fast response to changes in intracellular T3 concentration, especially in tissue repair. In addition, the T3 concentration is related to physical conditions such as cold exposure by increasing AMP in the brown adipocytes and increasing the T4 to T3 conversion by DIO2 mediation, thus leading to adaptive thermogenesis by uncoupling protein 1 [175]. DIO2 plays a role in regulating the hypothalamus–pituitary–thyroid axis by controlling TSH secretion caused by the intracellular concentration of T3 [176]. Moreover, the overexpression of DIO2 could affect the cell cycle. It arrests the trophoblast cell line proliferation at the G1 phase by downregulating the proliferating cell nuclear antigen and cyclin-D1. It stimulates apoptosis via inhibition of AKT and the increase of caspase-3 activity (Adu-Gyamfi, 2021). 

## 6. Selenoproteins and Their Polymorphisms 

Selenoproteins’ polymorphism has been reported as an important genetic factor that generates functional consequences and correlations with various diseases, especially cancer and cardiovascular diseases.

### 6.1. Glutathione Peroxidases

#### 6.1.1. rs1050450 Polymorphism

Polymorphisms could influence the concentration of Se and even the protein structure and functionality. One of the most researched single nucleotide polymorphisms (SNP) is related to GPX1 in rs1050450 (position 198), which is a C to T substitution in exon 2 resulting in an amino acid change from proline (Pro) to leucine (Leu), which affects the protein function [177]. This polymorphism was also analyzed in Brazilian Alzheimer’s patients, and the results indicate the association between the erythrocyte Se concentration and GPx activity affecting the protein function in those patients [51]. In addition, the TT homozygote patients of this polymorphism had lower scores of long-term visual memories in Alzheimer’s disease compared to the CC and CT genotypes in a cohort of 334 individuals in Brazil [52]. This polymorphism was also studied in another mental disorder. The carriers of C/T haplotypes of this polymorphism have a higher susceptibility to schizophrenia in the Chinese Han population [93]. On the other hand, no significant association was found between the Pro198Leu polymorphism and panic disorder in the panic and agoraphobia scale and age of onset [82].

Cancer is one of the most researched diseases concerning selenoprotein polymorphism associations. A study with 975 Danish patients showed that the Leu carriers for rs1050450 (Pro198Leu) polymorphism in GPX1 have a 1.9-fold increased risk of developing non-ductal breast cancer and are 2.6-fold more likely to have a grade 3 ductal tumor compared to a grade 1 or 2 [178]. Another study with 377 breast cancer patients revealed that the carriers of the T allele of the Pro198Leu polymorphism had a 1.43-fold higher risk of breast cancer compared with non-carriers among Danish women. In contrast, other groups failed to find an association between the rs1050450 polymorphism and breast cancer in patients from Rwanda, the United States, Canada, and nine other well-established cohort studies [58,59,60,61,87]. Associations were also demonstrated in lung cancer [78]. In a South Korean cohort, the codon 198 polymorphism carriers with the Pro/Leu or Leu/Leu genotype had a higher risk for lung cancer and higher urinary 8-oxoguanine glycosylase 1 concentrations compared to those with the Pro/Pro genotype [78]. The individuals with one or two variant alleles of the same polymorphism showed an increased risk of lung cancer compared to wild-type individuals. The variants were prevalent among middle-aged Finnish smokers. The group interpreted this as a possible explanation for the existence of a less efficient glutathione peroxidase complex caused by the polymorphism [79]. 

In addition, some studies investigated bladder cancer. A study in the Japanese population showed that the rs1050450 Pro/Leu genotype was highly associated with advanced tumor stage and increased the risk of bladder cancer compared with the control group [57]. The Ecuador sub-population that carries this polymorphism presented 3.8 times greater probability of developing bladder cancer than controls [53]. In a Serbian cohort with 330 patients with urothelial bladder cancer, the rs1050450 polymorphism was not associated with the risk of the disease but with the aggressiveness of the tumor stage and pathohistological grade [55]. In contrast, a protective association was found in a cohort of 224 subjects. Patients with the wild Pro198Leu polymorphism had shorter recurrence-free survival for superficial bladder cancer than the variant genotype [56]. No associations were confirmed in the Morocco population [54].

In the Turkish population, at position 198 of the GPX1 gene, the samples with the Pro/Leu or Leu/Leu genotypes of the polymorphism showed a higher risk of prostate cancer. The Leu/Leu allele was more frequent in patients with high-stage disease than those with the lower stage, suggesting an association with the development and progression of the disease [85]. Another research group with a small cohort in Turkey found no associations of this polymorphism between prostate cancer’s aggressive and non-aggressive severity [86]. A protective effect of the Leu variant was found in the North Macedonia population, where the carriers of this variant allele had a lower risk of prostate cancer than wild-type individuals [88]. No associations were found between the rs1050450 polymorphism and other cancers, such as endometrial cancer in Polish women [71], papillary thyroid carcinoma in the Iran population [92], colorectal adenomas and colorectal cancer in Norwegian patients [64], and laryngeal cancer survival patients in the Poland population [83]. 

Several studies highlighted the involvement between the GPX1 rs1050450 polymorphism and other diseases. In the Egyptian population, the T allele of the polymorphism was highly associated with keratoconus risk. This allele is widespread among the severe stages compared to moderate and mild ones, suggesting a correlation with the development and progression of keratoconus. The C/C genotype was associated with protection against the disease [179]. Similar results were found in an Iran cohort [75] and the Turkish population [76]. In the Polish cohort, the T/T genotype was associated with the progression of open-angle glaucoma, and the C/T genotype carriers had an increased risk for the disease [73]. Studies also aimed to evaluate the association between this variation and diabetes mellitus. In a cohort with 1244 type 2 Polish patients, the T allele and T/T genotype increased the risk of developing diabetic peripheral neuropathy [68]. Similar results were found in the Turkish population, in which the T allele was significantly more frequent in patients with new onset diabetes mellitus after renal transplantation [180]. When the evaluation of macrovascular diseases in type 2 diabetic Japanese patients was performed, the prevalence of peripheral vascular and cardiovascular diseases and the mean intima-media thickness of the common carotid arteries was higher in the Pro/Leu group when compared to the Pro/Pro group [181]. In the Mexican population, authors could not find an association between obese patients with prediabetes or diabetes and the rs10504050 polymorphism, but it was associated with obesity in nondiabetic participants [182]. 

Pro198Leu was associated with a deficiency in blood concentration of Se and lower GPx1 activity in patients with Keshan disease, a Se deficiency endemic in China. In cultured cardiomyocytes from neonatal rats, the overexpression of the leucine polymorphism caused a 30% reduction in GPX 1 activity and increased apoptosis induced by serum starvation compared to wild-type variants [62]. The Pro/Leu and Leu/Leu genotypes of Pro198Leu polymorphisms were also significantly associated with a risk of Kashin–Beck disease, an osteoarthropathy endemic to regions with Se deficiency in China. Additionally, the blood’s GPX activity was lower in the Kashin–Beck group with Pro/Leu and Leu/Leu genotypes [74]. Nephrolithiasis patients had a higher frequency of the genotype carrying at least the Leu allele than controls in Iran, suggesting an elevated risk for the disease development [80].

rs1050450 may influence longevity in humans. A Danish cohort with 1905 individuals found a decreased mortality of carriers with T alleles and increased Activity of Daily Living scores [77]. The risk of restenosis after coronary artery stenting with bare metal stents was evaluated, and results showed that carriers of the T allele were associated with the risk of in-stent restenosis, a decrease of the protein activity, and higher lipoperoxides in LDL in 74% [90]. The Pro198Leu CT genotype had a 2.84-fold risk for coronary artery disease and low erythrocyte GPX1 activity with increased severity in all score groups, suggesting that this substitution is less responsive in stimulation of the protein activity in those patients in Sri Lanka [66]. Preeclampsia was also associated with a risk factor for rs1050450 polymorphism, showing a 1.7- to 1.6-fold increased risk for the disease in Iranian women [89]. The carriers of the Leu allele of rs1050450 had higher levels of inflammatory fibrinogen, especially D-dimmer in COVID-19 patients from Serbia [183].

A lack of associations was also found between the rs1050450 polymorphism and spontaneous abortion in Iran women [49], coronary artery disease in Taiwanese and Tunisian populations [65,67], distal symmetric in Polish patients with type 2 diabetes mellitus [69], peripheral neuropathy in type 2 diabetic patients in Pakistan [70], cataract in Chinese population [63], idiopathic male infertility in Iran [184], fibromyalgia in the Turkish population [72], nonalcoholic steatohepatitis in the Taiwanese population [91], and mercury levels in mildly exposed women from Brazilian Amazon [50]. 

#### 6.1.2. Alanine Repetition Polymorphism

Other types of polymorphisms have been investigated in GPX1. The variable number of GCG repetitions, between four and six, can show differences between diverse diseases. In breast cancer, the allele containing four GCG was associated with the disease risk in premenopausal women in Canada [61]. Variation at position 337 of exon 1 results in a variable amount of alanine (ALA) in the polyalanine (142) coding sequence, ranging between 5 and 7 [185]. This repetition affects the modulation of the gene expression in response to Se [186]. In in vitro analyses, the combination of Ala6 (six repetitions) and rs1050450 (Pro-Leu) had a 40% decrease in enzyme activity [181]. A study with 61 patients with index squamous cell carcinoma who developed a second primary tumor in the United Kingdom showed a significant association where the patients had increased chances of possessing the ALA (seven repetitions) allele compared with the population control [47]. On the other hand, no significant associations were found in the carriers of the GCG polymorphism in prostate cancer patients in the United Kingdom. However, an increased frequency of Ala6 was observed in patients compared to controls [48]. Five ALA alleles were more frequent in autism spectrum disorder patients. When analyzed in vitro, GPx1 had lower activity in the ALA5 polymorphism [45]. The ALA6 polymorphism may have protective factors for autism spectrum disorder due to a lower transmission of the polymorphism through parents in the disease [46].

#### 6.1.3. rs1800668 Polymorphism

The other relevant polymorphism in GPX1 is rs1800668, which changes the C to a T nucleotide located in the gene’s promoter region. A study in Iran with 158 subjects showed that the protein activity was higher among the CC homozygotes than in other participants, but this variation did not alter the protein structure [187]. In the New Zealand population, the polymorphism was associated with an increased risk of first diagnosis of Crohn’s disease in individuals before 17 and between 17 and 40 years old. The location of the disease had an increased risk of ileal disease [95]. In addition, this variation was associated with cumulative lead exposure and meningioma [94]. As observed above, few studies compared this polymorphism to a particular disease.

#### 6.1.4. GPx3 Polymorphism

Even though the GPx3 protein is commonly used as a biomarker for Se status, few polymorphisms have been studied. In the Mexican population, the rs8177409 SNP (also called -302 A/T) was associated with cardiovascular risk, and the protein level was increased in patients with metabolic syndrome [97]. In the Taiwanese population, the G allele of the rs3805435 SNP protected against sudden sensorineural hearing loss. However, the group did not find an association with rs3763013, rs8177412, rs3828599, and rs2070593 SNPs [98]. In contrast, the rs8177412 genotype was significantly associated with the risk of developing testicular germ cell tumor and seminoma in Serbian men [99]. In the Chinese population, the risk of schizophrenia did not correlate with the rs736775 polymorphism between cases and controls [96].

#### 6.1.5. GPx4 Polymorphism

The rs713041 polymorphism in position 718 is the most researched variant in the glutathione peroxidase 4 gene, causing a T to C substitution. This alteration in the 3’UTR altered the protein binding to the 3’UTR and influenced the concentration of GPx4 and other selenoproteins [188]. The overexpression of the rs713041 variant altered the protein expression and viability of Caco-2 cells [189]. It influenced the adhesion events in vascular cells, affecting cardiovascular health and diseases. Endothelium cells with the T allele were more sensitive to oxidative stress and had increased levels of lipid hydroperoxides than the C allele [102]. This polymorphism seemed to modulate negatively eGPX activity in T/T carriers who later developed breast cancer compared to those who did not develop the disease [190]. It also impacted the risk for cardiac autonomic neuropathy in type 1 diabetes patients in Brazil [106]. The carriers of the T/T genotype for rs713041 had a higher risk of acute pancreatitis recurrence, and it was associated with increased malonyldialdehyde concentrations, a marker for oxidative stress induced by inflammation [81]. In this study, a cohort of 667 hypertensive patients showed that the C718T polymorphism was associated with an increased risk of cerebral stroke, suggesting a probable susceptibility to the disease in the Russian population. On the other hand, the combination of GPX1, GPX3, and GPX4 variants did not show a significant association [103]. It has also been demonstrated an association between the polymorphism and preeclampsia in the Chinese population [100,105] and with the development and severity of endometriosis in Taiwanese women [104]. However, male infertility in Italian men was not associated with this variant [191]. The other two polymorphisms, rs4965814 and rs9874, conferred risk for ischemic stroke only in women [101]. The rs713041, rs2074451, and rs3746165 polymorphisms did not demonstrate an association with autoimmune thyroid diseases in the Chinese population [107]. 

### 6.2. Selenoprotein P

Regarding selenoprotein P, two of the most researched polymorphisms, rs7579 and rs3877899, affect biomarkers of selenoprotein status and disease susceptibility. The first variant, rs7579, changes a G to an A at position 25191, and the second variant, rs3877899 or Ala234Thr, changes the amino acid alanine to a threonine by the G/A variant at position 24731 [190]. Compared to the common allele, the rs7579 genotype affects the protein serum concentration in the less frequent allele of the heterozygous and homozygous carriers. The A/A carriers showed an increased risk of prostate cancer in the German population [84]. The A allele of rs7579 was associated with higher odds of metabolic syndrome in a Chinese cohort where both G/A and A/A genotype carriers had increased association of odds ratios for the disease than the G/G genotype [125]. Individuals carrying the A/A allele of rs3877899 had a 3.8-fold higher risk of breast cancer compared with the G/G allele. The carriers of the heterozygous genotype also had an increased risk for the disease in Iranian women [126]. The rs3877899 polymorphism was also associated with the risk of treatment failure of prostate cancer within the first two years after surgery in African-American men, suggesting higher chances of recurrence, and the serum selenium levels are lower in the same group when compared to Caucasian subjects [35]. A study showed that two other different polymorphisms were associated with prostate cancer: the carriers of the T allele for the rs11959466 variation had an increased risk for the disease, while the rs13168440 carriers with the minor allele had a decreased risk in the United States [127]; carriers of the rs13154178 polymorphism had higher levels of fasting blood glucose in pregnant Turkish women, suggesting insulin resistance in gestational diabetes mellitus [129]. The rs230820 polymorphism was not associated with the risk of aggressive prostate cancer at diagnosis in male citizens of the United States [128]. The rs6865453 polymorphism of this selenoprotein was not associated with autoimmune thyroid diseases in a cohort of 1060 Chinese patients [107]. In addition, no association was found between the r25191g/a polymorphism and Kashin–Beck disease in a cohort of 167 cases in China. This polymorphism is 25191 in the 3’ untranslated region [44].

### 6.3. Selenoprotein F

The selenoprotein F gene also has two significant single nucleotide polymorphisms, one at position 811 (rs5845 C/T) and the other at 1125 (rs5859 G/A) in the 3’ untranslated region in humans, and they could result in different responses to Se supplementation and change the selenocysteine incorporation into the protein [192,193]. The frequency of rs5859 polymorphism is higher in Kashin–Beck patients when compared to healthy controls in the Chinese population, and the A allele is linked to an increased risk of the disease, suggesting that this variant could lead to uncommon expression of the protein and lead to apoptosis in those patients [115]. In a cohort with 139 HIV-a positive patients, A/A homozygotes of the same polymorphism were associated with a shorter progression to AIDS than the G/G genotype, suggesting a possible risk biomarker [120]. The carriers of the A/A genotype of rs5859 SNP had a 2.8-fold higher risk of breast cancer than the G/G genotype in Iranian women [126]. In the South Korean population, the minor alleles for rs5845 and rs5859 were associated with an increased risk of rectal cancer in men in a cohort of 827 patients [119]. No associations were found between the rs5845 polymorphism and breast cancer in Caucasian women in Austria [194]. The 1125 G/A polymorphism could modulate the influence of dietary Se in lung cancer patients from Poland. In cultured cells, this polymorphism was associated with a lower response to added Se [195].

### 6.4. Selenoprotein S

Studies in SelS’s polymorphism are also linked to the intermediation of the inflammatory response. The -105G>A polymorphism is associated with the plasma levels of IL1 beta, IL6, and TNF-alpha, significantly when suppressed by interfering RNA in macrophage cells, increasing the ranks of the cytokines and mediating the inflammatory response in a cohort of 522 individuals from the United States [196]. The carriers of the A allele of the same polymorphism increased the risk of gastric cancer of the intestinal type and the one located in the middle third of the stomach, influencing the inflammatory conditions of the mucosa in the Japanese population [121]. The -105 polymorphism is associated with higher susceptibility to spontaneous preterm birth with and without premature rupture of membranes and between extremely preterm neonates and controls in the Chinese population [122]. At least one A allele of this variation is also associated with Hashimoto’s thyroiditis in a Portuguese cohort where the male patients had 3.94 times increased risk for the disease [123]. Moreover, the carriers of the T variant for the rs34713741 SNP were associated with a higher risk of rectal cancer only in women [119]. The polymorphisms rs11327127, rs28665122, rs4965814, rs12917258, rs4965373, and rs2101171 did not have an association with autoimmune inflammatory diseases in the Spanish population [124].

### 6.5. Thioredoxin Reductase

TXNRD selenoprotein polymorphisms have not been well studied, but a few groups have tried to evaluate if there is an association with or between some diseases. One polymorphism in TXNRD1 was evaluated in advanced distal adenoma patients in the United States, and results have shown a significant reduction risk of 80% for carriers of the IVS1-181C>G allele at TXNRD1. This polymorphism is in a small gene nested inside the TXNRD1 intron, and it could change an SRY into an NkX-2 transcription factor binding site [114]. In Slovenia, a cohort of 972 patients with type 2 diabetes mellitus was analyzed, and this confirmed an association between the rs4485648 polymorphism of TXNRD2 and the disease. In summary, the CT heterozygotes had a 4.6-fold increased risk for diabetes mellitus, and the susceptibility of the illness was 4.3 times higher in subjects with the TT genotype than in the group with the CC genotype, which are essential data to propose a risk biomarker for the disease [116]. This polymorphism is located in intron 1 and suggests the possibility of contributing to the disease risk by splicing events and regulating gene expression [197]. The rs1005873 polymorphism in an intron of TXNRD2, which was related to the risk of aggressive prostate cancer at diagnosis in a cohort of 778 Americans. In addition, the patients who carried the polymorphism had a higher level of plasma Se, but the exact mechanism explaining this correlation is still unknown [117]. A few studies failed to demonstrate any associations with a specific disease, such as Kashin–Beck disease and the rs1139793, rs5748469, and rs5746841 polymorphisms of TXNRD2 in China [74,115,117,118]. 

### 6.6. Iodothyronine Deiodinases

The interaction between selenoprotein iodothyronine deiodinases (DIOs) and the activation and inactivation of the thyroid hormones by reductive deiodination has been elucidated with studies evaluating its polymorphisms. In a study with 387 Spanish mother–neonate, it was found that the intronic rs2235544 DIO1 polymorphism had negative associations between methylparaben, butylparaben, propylparaben, total triiodothyronine, bisphenol A, and free T4. The inverse association for bisphenol A and FT4 was found for rs12885300 CC and rs12885300 CT, and this was positive for rs12885300 TT in DIO2 [198]. The 3′UTR rs11206244 DIO1 polymorphism was associated with lifetime major depression in white females in a study performed in the United States. Moreover, the T allele was associated with increased free thyroxine levels in white and African-American subjects [108]. A group in the Netherlands identified a polymorphism in the most upstream short open reading frames (ORFa) of the 5’-UTR of DIO2, called D2-ORFa-Gly3Asp, and it changes the last codon from a glycine to an aspartic acid. In healthy donors, this polymorphism was associated with lower levels of plasma T4, free T4, and rT3 in a dose-dependent manner, but not in older adults, suggesting an age-dependent effect [199]. The −258A/G (rs12885300) DIO2 polymorphism was associated with a decreased rate of hormone secretion, such as TSH-stimulated free T4 secretion with a standard T3 release in patients from the United States [200]. A study with 269 Polish patients diagnosed with endometrial cancer revealed a 1.99-fold higher risk of developing endometrial cancer in CC homozygotes of the missense rs225014 DIO2 polymorphism compared to TT homozygotes [71]. 

In a cohort of 6,022 Korean participants, the Thr92Ala DIO2 polymorphism in women had a significantly lower axial speed of sound and T-score in the tibia than in control participants, suggesting an association between the polymorphism and the maintenance of bone mineral density, which could lead to the diagnosis of osteoporosis in women [201]. In addition, the heterozygous Thr92Ala DIO2 polymorphism was also associated with a 47% reduced risk of intrahospital mortality in adult Brazilian patients with COVID-19, suggesting a protective role of this polymorphism in the disease [111]. The rs945006 DIO3 minor G allele polymorphism was significantly more frequent in the poor outcome group compared with the excellent outcome group. The TT genotype was associated with a better outcome one year after acute ischemic stroke than one minor G allele in Lithuania patients [110]. Some studies could not find an association between the polymorphisms and certain diseases: Thr92Ala polymorphism and the severity of obesity after bariatric surgery in Italy [112], Thr92Ala polymorphism and cognitive impairment in older adults in Brazil [113], rs12885300 polymorphism and the body weight variation after Graves’ disease treatment in Brazil [109], and rs12095080, rs11206244, rs2235544 polymorphisms in DIO 1 and rs225014 and rs225015 in DIO2 with acute ischemic stroke outcome [110].

### 6.7. Selenophosphate Synthetase

Finally, although the role of selenophosphate synthetase is uncertain, it is known that this enzyme catalyzes the conversion of selenide to selenophosphate, the Se component required for selenocysteine and selenoprotein synthesis [202,203]. In mammals, the role of SEPHS2 is to provide the active Se donor, and SPS1 does not have a clear function established, but it could have a non-essential role in selenoprotein metabolism [204]. SEPHS2 is related to the aggressiveness and malignant tumor grade [205,206]. Its expression is related to immune infiltration, and it could act in the development of tumors through the peroxisome proliferator-activated receptors signaling pathway [205].

## 7. Conclusions

Selenoproteins have essential effects on inflammation, reduction of oxidative stress, thyroid hormone regulation, and even fertility. Different cellular and molecular mechanisms are affected by its action. Intake of Se in daily diet is recommended to maintain the homeostasis of metabolism. Still, studies have shown that the deficiency of Se could result in dysregulation of diverse mechanisms and susceptibility to diseases. One of the reasons for this failure in the distribution or intake of Se is the selenoproteins´ polymorphism. The variants of each selenoprotein can compromise the protein function of the levels of the organic form of Se and increase the risk for several diseases. These data are essential to map the genetic background within the disease distribution and risk assessment outcomes. In the future, understanding such mapping will possibly foster the development of molecular markers for diagnosis or even the development of personalized approaches based on specific forms of treatment for patients with a particular polymorphism, preventing possible disease progressions and deaths. This action will provide valuable insights into the disease mechanisms. Furthermore, through personalized medicine approaches, we can suggest that individuals carrying specific selenoprotein polymorphisms may be at risk or susceptible to developing some diseases (Table 2). However, scarce studies on this subject leave uncertainties about the action of these variations, mainly due to the external genetic influences of each country that can also modify its functionality. In addition, as observed, some selenoproteins lack current studies on their performance and characterization, which are the basis for a better understanding of how the protein and its polymorphisms act in homeostasis and various pathophysiologies. Lastly, this review proves that the study of Se and selenoproteins still has a long way to go. Several analyses have demonstrated the importance of its polymorphism as a risk biomarker in different diseases.

## Figures and Tables

**Figure 1 ijms-25-01402-f001:**
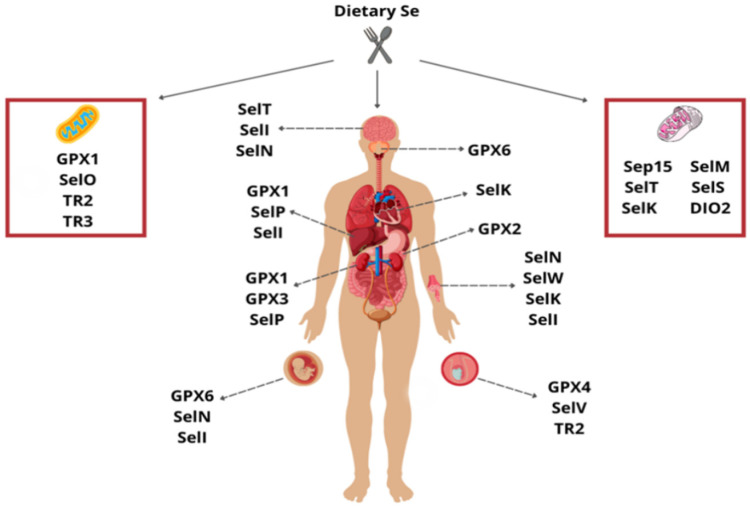
Se is mostly absorbed from diet intake. The liver is the main organ for Se regulation. It synthesizes selenoprotein P to maintain cellular Se homeostasis by its transportation to several tissues. It is understood that some selenoproteins can act in different organs and locations, as shown in the figure. From the glutathione peroxidase family, GPX1 is highly expressed in the liver and kidney and can be found in the mitochondria; GPX2 is predominantly expressed in the gastrointestinal epithelium; GPX3 is found in extracellular compartments, and it is mainly in the kidney; GPX4 is expressed in the testis; GPX6 is located in the olfactory epithelium and embryonic tissues. SelN is primarily expressed in the brain, skeletal muscle, and placenta, concerning other selenoproteins. SelW predominates in skeletal muscle; SelV and TXNR3 are expressed in the testis; DIO2, SelS, SelF, SelT, and SelK are in the endoplasmic reticulum, where SelK is especially found in the heart and skeletal muscle; SelI is mainly found in placenta, liver, and brain; TXNR2 and SelO are found in mitochondria. It is essential to understand that some selenoproteins have multiple subcellular locations, and the areas described in this figure only summarize where is located the highest expression of each protein.

**Table 1 ijms-25-01402-t001:** Selenoproteins identified in mammals and their data: exon number, functions, and location. The National Center for Biotechnology Information (NCBI) has all the genomic details. GPX: glutathione peroxidase, DIO: iodothyronine deiodinase, SEPHS2: selenophosphate synthetase 2, TXNRD: thioredoxin reductase, HPT: hypothalamus–pituitary–thyroid.

Selenoprotein Genes	Chromosome	Exons	Function	Ref.
SelH	11q12.1	4	Unknown	Mehta, 2013 [16]
GPX6	6p22.1	5	Reduction of hydroperoxides	Kryukov, 2003 [17]
GPX1	3p21.31	2	Reduction of hydroperoxides	Esworthy, 1997 [18]
GPX2	14q23.3	4	Reduction of hydroperoxides	Roman, 2014 [20]
GPX3	5q33.1	6	Reduction of hydroperoxides	Takahashi, 1986 [21]
GPX4	19p13.3	8	Reduction of oxidized phospholipids	Cozza, 2017 [25]
SelP	5p12	7	Sel transportation	Schomburg, 2022 [27]
SelF	1p22.3	6	Protein folding control	Gladyshev, 1998 [28] Labunskyy, 2007 [29]
SelS	15q26.3	7	Production of inflammatory cytokines	Gao, 2006 [30]
SelN	1p36.11	13	Unknown	Taylor, 2009 [31]
SelW	19q13.33	6	Unknown	Vendeland, 1995 [32]
SelV	19q13.2	6	Regulation of O-GlcNAcylation	Mariotti, 2017 [33]
SelT	3q25.1	6	Cell adhesion	Sengupta, 2009 [34]
SelI			Motor neuron function	Horibata, 2007 [35]
SelM	22q12.2	5	Protection of oxidative stress	Reeves, 2010 [36]
SelR	16p13.3	4	Regulation of methionine by methionine sulfoxide catalysis	Dai, 2016 [37]
SelO	22q13.33	9	Redox homeostasis	Han, 2014 [38]
TXNRD1	12q23.3	18	Regulatory mechanisms	Su, 2004 [39]
TXNRD2	22q11.21	22	Regulatory mechanisms	Su, 2004 [39]
SelK	3p21.1	5	Cofactor in protein palmitoylation	Fredericks, 2015 [40]
TXNRD3	3q21.3	16	Sperm maturation	Wang, 2022 [41]
DIO1	1p32.3	5	Thyroid hormone regulation by converting T4 to T3	Jakobs, 1997 [42]
DIO2	14q31.1	7	Regulation of the HPT axis	Baqui, 2003 [43]
DIO3	14q32.31	1	Inactivation of thyroid hormone	Baqui, 2003 [43]
SEPHS2	16p11.2	1	Active selenium donor	Copeland, 2000 [44]

**Table 2 ijms-25-01402-t002:** Selenoprotein polymorphisms and their associations with diseases. These polymorphisms could affect disease susceptibility or risk and disease progression to more severe clinical cases and/or provide carriers with protection against disease development.

Gene	Polymorphism	Country	Disease	Correlation with the Disease	References
GPX1	ALA repetition	Italy	Autism	Susceptibility/risk	Carducci, 2022 [45]
		United States	Autism	Protection	Xue Ming, 2010 [46]
		United Kingdom	Head and neck cancer	Progression	Jefferies, 2005 [47]
		United Kingdom	Prostate cancer	No correlation found	Z Kote-Jarai, 2002 [48]
	Pro198Leu (rs1050450)	Iran	Abortion	No correlation found	Eskafi Sabet, 2014 [49]
		Brazil	Alzheimer’s	No correlation found	Rocha, 2016 [50]
		Brazil	Alzheimer’s	Low Se concentration	Cardoso, 2012 [51]
		Brazil	Alzheimer’s	Susceptibility/risk	Da Rocha, 2018 [52]
		Japan	Bladder cancer	Susceptibility/risk	Ichimura, 2004 [53]
		Morocco	Bladder cancer	No correlation found	Hadami, 2016 [54]
		Ecuador	Bladder cancer	Susceptibility/risk	Paz-y-Miño, 2010 [55]
		United States	Bladder cancer	Protection	Zhao H, 2005 [56]
		Serbia	Bladder cancer	Susceptibility/risk	Nikic, 2023 [57]
		Rwanda	Breast cancer	No correlation found	Habyarimana, 2018 [58]
		United States	Breast cancer	No correlation found	Ahn, 2005 [59]
		United States	Breast cancer	No correlation found	Cox, 2004 [60]
		Canada	Breast cancer	No correlation found	Knight, 2004 [61]
		Poland	Breast cancer	Protection	Jablonska, 2015
		China	Cardiomyopathy	Susceptibility/risk	Lei C, 2009 [62]
		China	Cataract	No correlation found	Zhang, 2011 [63]
		Norway	Colorectal cancer	No correlation found	Hansen, 2005 [64]
		Taiwan	Coronary disease	No correlation found	Hseng-Long Yeh, 2018 [65]
		Sri Lanka	Coronary disease	Susceptibility/risk	Wickremasinghe, 2016 [66]
		Tunisia	Coronary disease	No correlation found	Souiden, 2016 [67]
		Serbia	COVID-19	No correlation found	Jerotic, 2022
		Poland	Diabetes	Susceptibility/risk	Buraczynska, 2017 [68]
		Poland	Diabetes	No correlation found	Kasznicki, 2016 [69]
		Pakistan	Diabetes	No correlation found	Mushtaq, 2020 [70]
		Turkey	Diabetes	Susceptibility/risk	Yalin, 2017 [12]
		Poland	Endometrial cancer	No correlation found	Janowska, 2022 [71]
		Turkey	Fibromyalgia	No correlation found	Akbas, 2014 [72]
		Poland	Glaucoma	Susceptibility/risk	Malinowska, 2016 [73]
		China	Kashin–Beck disease	Susceptibility/risk	Xiong, 2010 [74]
		Iran	Keratoconus	Susceptibility/risk	Yari, 2018 [75]
		Turkey	Keratoconus	Susceptibility/risk	Abdullah Ilhan, 2019 [76]
		Denmark	Longevity	Protection	Soerensen, 2009 [77]
		South Korea	Lung cancer	Susceptibility/risk	Chul-Ho Lee, 2006 [78]
		Finland	Lung cancer	Susceptibility/risk	D Ratnasinghe, 2000 [79]
		Iran	Nephrolithiasis	Susceptibility/risk	Aghakhani, 2017 [80]
		Poland	Pancreatitis	Susceptibility/risk	Ściskalska, 2022 [81]
		Turkey	Panic syndrome	No correlation found	Cengiz, 2015 [82]
		Poland	Pharyngeal cancer	No correlation found	Lubiński, 2021 [83]
		Germany	Prostate cancer	Susceptibility/risk	Astrid Steinbrecher, 2010 [84]
		Turkey	Prostate cancer	Progression	Kucukgergin, 2011 [85]
		Turkey	Prostate cancer	No correlation found	Erdem O, 2012 [86]
		Denmark, France, Great Britain, Germany, Greece, Italy, Netherlands, Spain, and Sweden	Prostate cancer	No correlation found	Blein, 2014 [87]
		Macedonia	Prostate cancer	Protection	Arsova-Sarafinovska, 2009 [88]
		Iran	Pulmonary embolism	Susceptibility/risk	Teimoori, 2019 [89]
		Russia	Restenosis	Susceptibility/risk	Shuvalova, 2012 [90]
		Taiwan	Steate hepatitis	No correlation found	Huang, 2021 [91]
		Iran	Thyroid carcinoma	No correlation found	Salimi, 2020 [92]
	rs1800668 and rs1050450	China	Schizophrenia	Susceptibility/risk	Xiaojun Shao, 2020 [93]
	rs1050450 and rs18006688	United States	Brain tumor	Susceptibility/risk	Bhatti, 2009 [94]
	rs1800668	New Zealand	Crohn’s disease	Susceptibility/risk	Morgan, 2010 [95]
		China	Schizophrenia	Protection	Xiaojun Shao, 2020 [93]
GPX3	rs736775	China	Macrovascular disease	No correlation found	Liu C, 2020 [96]
	rs8177409	Mexico	Metabolic syndrome	Susceptibility/risk	Baez-Duarte, 2014 [97]
	rs3805435	Taiwan	Hearing loss	Protection	Chien, 2017 [98]
	rs3763013, rs8177412, rs3828599, and rs2070593	Taiwan	Hearing loss	No correlation found	Chien, 2017 [98]
	rs8177412	Serbia	Testicular tumor	Susceptibility/risk	Bumbasirevic, 2022 [99]
GPX4		China	Preeclampsia	No correlation found	Peng, 2016 [100]
	rs4965814 and rs9874	Finland	Cardiovascular disease	Susceptibility/risk	Silander, 2008 [101]
	rs713041	Scotland	Vascular disease	Susceptibility/risk	Crosley, 2013 [102]
		Russia	Stroke	Susceptibility/risk	Polonikov, 2012 [103]
		Taiwan	Endometriosis	Progression	Huang, 2020 [104]
		Poland	Pancreatitis	Susceptibility/risk	Ściskalska, 2022 [81]
		China	Preeclampsia	Susceptibility/risk	Chen, 2020 [105]
		China	Preeclampsia	Susceptibility/risk	Peng, 2016 [100]
		Brazil	Diabetes	Protection	Admoni, 2019 [106]
	rs713041, rs2074451, and rs3746165	China	Thyroid disease	No correlation found	Xiao, 2017 [107]
DIO1	rs11206244	United States	Depression	Susceptibility/risk	Philibert, 2011 [108]
DIO2	rs12885300	Brazil	Graves’ disease	Susceptibility/risk	Comarella, 2021 [109]
	rs225014 and rs225015	Lithuania	Stroke	No correlation found	Taroza, 2019 [110]
	Thr92Ala [rs225014]	Brazil	COVID-19	Protection	Beltrão, 2022 [111]
		Italy	Obesity	No correlation found	Benenati, 2022 [112]
		Brazil	Cognitive impairment	No correlation found	Schwengber, 2022 [113]
DIO3	rs945006	Lithuania	Stroke	Progression	Taroza, 2019 [110]
	rs12095080, rs11206244, and rs2235544	Lithuania	Stroke	No correlation found	Taroza, 2019 [110]
TXNRD1	rs35009941	United States	Colorectal cancer	Protection	Peters, 2008 [114]
TXNRD2	rs1139793	China	Kashin–Beck	No correlation found	Wu, 2019 [115]
	rs4485648	Slovenia	Diabetes	Susceptibility/risk	Kariž, 2015 [116]
	rs5748469	China	Kashin–Beck	No correlation found	Lu, 2010 [117]
	rs5746841	China	Kashin–Beck	No correlation found	Li, 2019 [118]
SelF	1125 G/A	Poland	Lung cancer	Modifies the selenium level	Jablonska, 2015
	rs5845 and rs5859	South Korea	Colorectal cancer	Susceptibility/risk	Sutherland, 2010 [119]
	rs5859	China	Kashin–Beck	Susceptibility/risk	Wu, 2019 [115]
	rs5859	Brazil	AIDS	Progression	Benelli, 2016 [120]
SelS	rs34713741	South Korea	Colorectal cancer	Susceptibility/risk	Sutherland, 2010 [119]
	-105G --> A	Germany	Inflammatory bowel disease	No correlation found	Seiderer, 2007
	-105G --> A	Japan	Gastric cancer	Susceptibility/risk	Tomoyuki, 2009 [121]
	-105G>A	China	Premature birth	Susceptibility/risk	Wang, 2013 [122]
	-105G>A	Portugal	Hashimoto’s thyroiditis	Susceptibility/risk	Santos, 2014 [123]
	rs11327127, rs28665122, rs4965814, rs12917258, rs4965373, and rs2101171	Spain	Autoimmune inflammatory disease	No correlation found	Martínez, 2008 [124]
SelP	rs7579	China	Metabolic syndrome	Susceptibility/risk	Li Zhou, 2020 [125]
	rs3877899	United States	Breast cancer	Susceptibility/risk	Mohammaddoust, 2018 [126]
	rs11959466	United States	Prostate cancer	Susceptibility/risk	Penney, 2013 [127]
	rs13168440	United States	Prostate cancer	Protection	Penney, 2013 [127]
	rs230820	United States	Prostate cancer	No correlation found	Xie, 2016 [128]
	r25191G/A	China	Kashin–Beck	No correlation found	Sun, 2010 [44]
	rs7579	Germany	Prostate cancer	Susceptibility/risk	Astrid Steinbrecher, 2010 [84]
	rs146125471, rs28919926, and rs16872762	Turkey	Diabetes	No correlation found	Akbaba, 2018 [129]
	rs13154178	Turkey	Diabetes	Susceptibility/risk	Akbaba, 2018 [129]
	rs7579	Poland	Endothelial cancer	No correlation found	Janowska, 2022 [71]

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
