# Peer review of "Current Understanding of Human Polymorphism in Selenoprotein Genes: A Review of Its Significance as a Risk Biomarker"

_ijms, 2024, doi:10.3390/ijms25031402_

Round 1

Reviewer 1 Report

Comments and Suggestions for Authors

The manuscript attempts to review the current literature on selenoprotein gene polymorphisms and how theses SNPs can become biomarkers for different diseases. Despite its length, the review should be rewritten and reorganized. The topic is very interesting and necessary, as a comprehensive review on human selenoprotein gene SNPs is not currently available in the literature. However, by entering into inuendos on selenoproteins themselves, the manuscript becomes scattered and looses value for the scientific literature. To provide a novel perspective to the scientific community, the manuscript should focus on the SNPs without too much inuendos on selenoproteins, and even go further and target SNPs for specific diseases/conditions – being cancer and immune function the obvious choices based on what the Authors wrote. This way, the review would be stronger, more clear, concise and precise. There is a concerning mix of cellular vs molecular functions for each selenoprotein, plus lack of citation for critical functions that have been uncovered for several selenoproteins in recent years. Also on that note, as a review, the manuscript must include more of the recent literature on the topic, as there has been significant breakthroughs in the last few years not mentioned in the text. As a recurrent issue, many sentences are too generic, filled with moot points, and they should be rephrased/removed to improve clarity upon re-writing of the manuscript.

MAJOR COMMENTS:

Title:

-       The word “human” should be present, as selenoprotein genes SNPs being reviewed here only include human ones.

Abstract:

-       Historical context: In the AbstractÊ» opening sentence, it is stated that since 1817, selenium is “proved to influence in several biological 12 functions, proving to be an essential micronutrient”. This is incorrect. The importance of selenium for health was only demonstrated in 1950Ê»s, as the Authors themselves state in the Introduction. From 1817 until 1950Ê»s, selenium was solely considered a toxin. Please rephrase the opening sentence to clarify the historical context.

-       L. 28 and 29: what kind of studies? “Some diseases” what diseases? These sentences need further specificity. Same with “certain diseases”, needs to be further identified.

-       Polymorphisms are in selenoprotein genes, reflected then in selenoproteins themselves. Please correct.

Introduction:

-       L. 67-68: selenoproteins do not “code for selenocysteine. Please rephrase.

-       As chromosome location is pointed in Table 1, the first column should be of selenoprotein genes using the current nomenclature, not selenoprotein themselves.

-       Table 1 suggests the involvement of selenoprotein in several functions, however specific molecular mechanisms have been clarified for some of them (e.g. ER misfolded protein machinery). The “function” column should be more specific. Also, there are a few typos in the “Function” column. SelR function: “redox homeostasis regulation of methionine” is too generic and should be much more specific as the selenoprotein formerly known as SelR has been widely studied. PMID 34167027 and PMID 32846216 suggest a function for SELENOV. What does “regulatory mechanisms” mean for TrxRs 1 and 2?

GPXs:

-       L. 82: what does “initiation transcription” mean in this context? Sentence seems broken.

-       The role of GPX4 in ferroptosis should be mentioned and thoroughly discussed.

Selenoproteins:

-       L. 103: “In humans, there are different types of Selenoproteins.” This is moot sentence. Please rephrase.

-       PMID 36516721 describes a molecular function for SELENOW that it should be mentioned.

-       This topic should be broken down into several paragraphs.

-       If SELENOK  “acts as a cofactor during the protein palmitoylation by binding to endoplasmic reticulum associated protein degradation components” why is it stated as “involved in calcium flux” in Table 1?

Iodothyronine Deiodinases:

-       L. 186-187: “Unfortunately, its function in the organism is not fully clear.” This is incorrect, and it should be updated with deiodinasesÊ» function.

Selenoprotein SNPs:

-       Some polymorphisms in the selenoproteins have been reported to have functional consequences and correlations with several diseases, especially cancer and cardiovascular diseases.” The sentence should be rephrased as it is very generic, as most significant polymorphisms in any gene affect function. Why specifically selenoprotein polymorphisms should be the focus of this introductory sentence.

-       Clearly GPX1 rs1050450 polymorphism is highly studied. Maybe it deserves its own section.

-       L. 483: the role of selenophosphate synthetase 1 (SEPHS1) is uncertain, not SEPHS2. Please rephrase.

Figures:

-       Fig 1 legend: “Se is MOSTLY absorbed from diet intake”.

-       Fig. 1 gives the impression that those selenoproteins are solely expressed in the pointed organs, which is incorrect. Particularly GPX4, DIO2 and TRXRÊ»s function in many other organs. Please rethink this figure layout.

MINOR COMMENTS:

-       L. 48: “mechanisms FOR RECODING”.

-       L. 58-59: Please rephrase. Correlation and associated in the same sentence are repetitive, and it should clarify direction of change (positive or inverse).

-       L. 119: “reticulUM”

-       L. 219-246: Please break down this paragraph.

-       A Table with main SNPs and their connection to diseases would add value to the manuscript.

Comments on the Quality of English Language

The manuscript should be further edited for conciseness and clarity, as there are several run-on and/or generic sentences. Grammar and syntax should be corrected.

Author Response

Dear Reviewer,

Thank you for sending the comments with recommended revision to the first version of our manuscript entitled “Current understanding of human polymorphism in selenoprotein genes: a review of its significance as risk biomarker”. According to the main comments, we made substantial changes in the manuscript and answered all comments, as you can find in detail below.

Thank you for the careful review process that improved a lot our manuscript.

Best regards,

Tania Araujo Jorge and Roberto Ferreira.

The manuscript attempts to review the current literature on selenoprotein gene polymorphisms and how theses SNPs can become biomarkers for different diseases. Despite its length, the review should be rewritten and reorganized. The topic is very interesting and necessary, as a comprehensive review on human selenoprotein gene SNPs is not currently available in the literature. However, by entering into inuendos on selenoproteins themselves, the manuscript becomes scattered and looses value for the scientific literature. To provide a novel perspective to the scientific community, the manuscript should focus on the SNPs without too much inuendos on selenoproteins, and even go further and target SNPs for specific diseases/conditions – being cancer and immune function the obvious choices based on what the Authors wrote. This way, the review would be stronger, more clear, concise and precise. There is a concerning mix of cellular vs molecular functions for each selenoprotein, plus lack of citation for critical functions that have been uncovered for several selenoproteins in recent years. Also on that note, as a review, the manuscript must include more of the recent literature on the topic, as there has been significant breakthroughs in the last few years not mentioned in the text. As a recurrent issue, many sentences are too generic, filled with moot points, and they should be rephrased/removed to improve clarity upon re-writing of the manuscript.

MAJOR COMMENTS:

Title:

-       The word “human” should be present, as selenoprotein genes SNPs being reviewed here only include human ones.

Response: Thanks for the comment. The word was included.

Abstract:

-       Historical context: In the AbstractÊ» opening sentence, it is stated that since 1817, selenium is “proved to influence in several biological 12 functions, proving to be an essential micronutrient”. This is incorrect. The importance of selenium for health was only demonstrated in 1950Ê»s, as the Authors themselves state in the Introduction. From 1817 until 1950Ê»s, selenium was solely considered a toxin. Please rephrase the opening sentence to clarify the historical context.

Response: Thanks for the right observation. We removed this information from the abstract and maintained it in the introduction.

-       L. 28 and 29: what kind of studies? “Some diseases” what diseases? These sentences need further specificity. Same with “certain diseases”, needs to be further identified.

Response: Thanks for the comment. We provided a more specific description.

-       Polymorphisms are in selenoprotein genes, reflected then in selenoproteins themselves. Please correct.

 Response: Thanks for the comment. We corrected it.

Introduction:

-       L. 67-68: selenoproteins do not “code for selenocysteine. Please rephrase.

 Response: Thanks for the comment. We corrected it.

-       As chromosome location is pointed in Table 1, the first column should be of selenoprotein genes using the current nomenclature, not selenoprotein themselves.

-       Table 1 suggests the involvement of selenoprotein in several functions, however specific molecular mechanisms have been clarified for some of them (e.g. ER misfolded protein machinery). The “function” column should be more specific. Also, there are a few typos in the “Function” column. SelR function: “redox homeostasis regulation of methionine” is too generic and should be much more specific as the selenoprotein formerly known as SelR has been widely studied. PMID 34167027 and PMID 32846216 suggest a function for SELENOV.

 Response: Thanks for the comment. We corrected it.

GPXs:

-       L. 82: what does “initiation transcription” mean in this context? Sentence seems broken.

-       The role of GPX4 in ferroptosis should be mentioned and thoroughly discussed.

 Response: Thanks for the comments. We discussed it.

Selenoproteins:

-       L. 103: “In humans, there are different types of Selenoproteins.” This is moot sentence. Please rephrase.

-       PMID 36516721 describes a molecular function for SELENOW that it should be mentioned.

-       This topic should be broken down into several paragraphs.

-       If SELENOK  “acts as a cofactor during the protein palmitoylation by binding to endoplasmic reticulum associated protein degradation components” why is it stated as “involved in calcium flux” in Table 1?

Response: Thanks for the comments. We discussed it more precisely and corrected these points.

Iodothyronine Deiodinases:

-       L. 186-187: “Unfortunately, its function in the organism is not fully clear.” This is incorrect, and it should be updated with deiodinasesÊ» function.

  Response: Thanks for the comment. We corrected it.

Selenoprotein SNPs:

-       “Some polymorphisms in the selenoproteins have been reported to have functional consequences and correlations with several diseases, especially cancer and cardiovascular diseases.” The sentence should be rephrased as it is very generic, as most significant polymorphisms in any gene affect function. Why specifically selenoprotein polymorphisms should be the focus of this introductory sentence.

-       Clearly GPX1 rs1050450 polymorphism is highly studied. Maybe it deserves its own section.

Response: Thanks for the comment. We provided a more specific description.

-       L. 483: the role of selenophosphate synthetase 1 (SEPHS1) is uncertain, not SEPHS2. Please rephrase.

Response: Thanks for the comment. We agreed with it.

Figures:

-       Fig 1 legend: “Se is MOSTLY absorbed from diet intake”.

-       Fig. 1 gives the impression that those selenoproteins are solely expressed in the pointed organs, which is incorrect. Particularly GPX4, DIO2 and TRXRÊ»s function in many other organs. Please rethink this figure layout.

Response: Thanks for the comment. We agreed with it. We improved the legend.

MINOR COMMENTS:

-       L. 48: “mechanisms FOR RECODING”.

-       L. 58-59: Please rephrase. Correlation and associated in the same sentence are repetitive, and it should clarify direction of change (positive or inverse).

-       L. 119: “reticulUM”

-       L. 219-246: Please break down this paragraph.

Response: Thanks for the comment. We addressed it.

-       A Table with main SNPs and their connection to diseases would add value to the manuscript.

Response: Thanks for the comment. We agree with the statement, but due to the short time for correction, we were unable to efficiently prepare this table.

Reviewer 2 Report

Comments and Suggestions for Authors

 Comments to the Authors of manuscript ID: ijms-2582498 entitled “Current understanding of genetic polymorphisms of selenoproteins as biomarkers for risk of diseases”.

Genetic polymorphisms in selenoproteins can serve as important biomarkers for assessing the risk of various diseases due to the critical role these proteins play in maintaining cellular function and antioxidant defense mechanisms. Authors highlighted that selenoproteins are essential components of the body's antioxidant defense system, playing roles in neutralizing ROS and maintaining redox balance. Adequate selenium levels and properly functioning selenoproteins are crucial for overall health.

Genetic variations in selenoprotein genes can lead to altered protein structure or function, affecting their antioxidant and enzymatic activities. Such variations have been linked to various diseases, including cancer, cardiovascular diseases, neurodegenerative disorders, and thyroid dysfunction. It is also mentioned.

But, it is not mentioned that glutathione peroxidases play a role in protecting blood vessels from oxidative stress and inflammation. Genetic variations that reduce their activity may contribute to increased cardiovascular risk. What about Parkinson disease and the potential role of selenoproteins in it?

1. L 27 – balance

2. L 28 – omitted “we describe”

3. L 57 – why only renal injury is mentioned? If the review focuses on the renal injury it is good option, but if Authors described many different health problems, other diseases please add

4. generally, introduction provided the basal information, gives the first look on the issue, but there is no hypothesis and the goal. It should be corrected.

5. The introduction ends with part about glutathione peroxidase 1, that should be moved into the next part. The table also.

6. there is a lack of the description of methods used during preparation of the text. What databases were used. The exclusion and inclusion criteria. All these should be described. What words were used for searching. How many texts were received initially, and how many were used finally. What was the range of time taking into account during searching.

7. L 370 – small letter

8. References are written in different font

9. in conclusion: it should be mentioned that understanding of individual`s genetic polymorphisms of selenoproteins can facilitate personalized medicine approaches, tailoring interventions based on genetic susceptibility to diseases. investigating the association between specific selenoprotein polymorphisms and disease risk provides valuable insights into disease mechanisms and potential preventive strategies. Identifying individuals with specific selenoprotein polymorphisms could aid in early disease detection, risk assessment, and personalized treatment strategies.

Author Response

Dear Reviewer,

Thank you for sending the comments with recommended revision to the first version of our manuscript entitled “Current understanding of human polymorphism in selenoprotein genes: a review of its significance as risk biomarker”. According to the main comments, we made substantial changes in the manuscript and answered all comments, as you can find in detail below.

Thank you for the careful review process that improved a lot our manuscript.

Best regards,

Tania Araujo Jorge and Roberto Ferreira.

Comments to the Authors of manuscript ID: ijms-2582498 entitled “Current understanding of genetic polymorphisms of selenoproteins as biomarkers for risk of diseases”.

Genetic polymorphisms in selenoproteins can serve as important biomarkers for assessing the risk of various diseases due to the critical role these proteins play in maintaining cellular function and antioxidant defense mechanisms. Authors highlighted that selenoproteins are essential components of the body's antioxidant defense system, playing roles in neutralizing ROS and maintaining redox balance. Adequate selenium levels and properly functioning selenoproteins are crucial for overall health.

Genetic variations in selenoprotein genes can lead to altered protein structure or function, affecting their antioxidant and enzymatic activities. Such variations have been linked to various diseases, including cancer, cardiovascular diseases, neurodegenerative disorders, and thyroid dysfunction. It is also mentioned.

But, it is not mentioned that glutathione peroxidases play a role in protecting blood vessels from oxidative stress and inflammation. Genetic variations that reduce their activity may contribute to increased cardiovascular risk. What about Parkinson disease and the potential role of selenoproteins in it?

Response: Thanks for the comment. We provided a more specific description.

  1. L 27 – balance
  2. L 28 – omitted “we describe”
  3. L 57 – why only renal injury is mentioned? If the review focuses on the renal injury it is good option, but if Authors described many different health problems, other diseases please add

Response: Thanks for the comment. We provided a more specific description.

  1. generally, introduction provided the basal information, gives the first look on the issue, but there is no hypothesis and the goal. It should be corrected. (Não sei o que colocar como hipotése até porque é uma revisão)

Response: Thanks for the comment. In the introduction section, we describe it and included more information.

  1. The introduction ends with part about glutathione peroxidase 1, that should be moved into the next part. The table also

Response: Thanks for the suggestion. We moved it.

.

  1. there is a lack of the description of methods used during preparation of the text. What databases were used. The exclusion and inclusion criteria. All these should be described. What words were used for searching. How many texts were received initially, and how many were used finally. What was the range of time taking into account during searching.

Response: Thanks for the suggestion. We included a description. Line 28.

  1. L 370 – small letter (Não entendi onde)
  2. References are written in different font

Response: Thanks for the comment. We corrected it.

  1. in conclusion: it should be mentioned that understanding of individual`s genetic polymorphisms of selenoproteins can facilitate personalized medicine approaches, tailoring interventions based on genetic susceptibility to diseases. investigating the association between specific selenoprotein polymorphisms and disease risk provides valuable insights into disease mechanisms and potential preventive strategies. Identifying individuals with specific selenoprotein polymorphisms could aid in early disease detection, risk assessment, and personalized treatment strategies

Response: Thanks for the comment. We improved the second version of our conclusion.

Reviewer 3 Report

Comments and Suggestions for Authors

the Authors present an overview of current knowledge on selenoproteins, with a focus on how their polymorphisms can influence the imbalance of physiological conditions.

While the review adds little - or nothing - to current knowledge in the field, it may have a good educational value.

I can offer the following comments:

- how evidence for inclusion was selected?

- section 6 is supposed to be the core of the manuscript and is, therefore, quite long. However, I think it would benefit from a reorganitation (dividing into subparagraphs?) and inclusion of some iconographic elements, figures or tables.

- can the Authors expand on how they see further studies on selenoproteins?

Author Response

Dear Reviewer,

Thank you for sending the comments with recommended revision to the first version of our manuscript entitled “Current understanding of human polymorphism in selenoprotein genes: a review of its significance as risk biomarker”. According to the main comments, we made substantial changes in the manuscript and answered all comments, as you can find in detail below.

Thank you for the careful review process that improved a lot our manuscript.

Best regards,

Tania Araujo Jorge and Roberto Ferreira.

the Authors present an overview of current knowledge on selenoproteins, with a focus on how their polymorphisms can influence the imbalance of physiological conditions.

While the review adds little - or nothing - to current knowledge in the field, it may have a good educational value.

I can offer the following comments:

- how evidence for inclusion was selected?

Response: Thanks for the comment. We provided a more specific description.

- section 6 is supposed to be the core of the manuscript and is, therefore, quite long. However, I think it would benefit from a reorganitation (dividing into subparagraphs?) and inclusion of some iconographic elements, figures or tables.

Response: Thanks for the comment. We improved the second version of our manuscript. We agree with the inclusion another table, but due to the short time for correction, we were unable to efficiently prepare it.

- can the Authors expand on how they see further studies on selenoproteins?   

Response: Thanks for the comment. In the conclusion section, we describe it and included more information.

Round 2

Reviewer 1 Report

Comments and Suggestions for Authors

The Authors have partially addressed key concerns but a few crucial corrections are needed (e.g. molecular function of deiodinases). Also, it is still not included a polymorphism table that two Reviewers seem to agree as necessary - maybe Authors can request a deadline extension so the Table can be prepared accordingly. In addition, it is puzzling why the selenoprotein nomenclature remains inconsistent, when they cite the 2016 paper in which the standardization of selenoprotein names was determined. Consistent use of selenoprotein names should be achieved, and removal of old names, such as SEP15 (but not restricted to this one). Table 1, for example, has old selenoprotein names instead of the standardized ones.

Comments on the Quality of English Language

The English improved from first version but still a few run-on sentences.

Author Response

Dear Reviewer,

Hereby, we would like to submit a revised version of our manuscript and a response to reviewer 1 letter. We greatly appreciate your help in revising this manuscript. The comments of reviewer 1 have been comprehensive and instructive. Reviewer 1 have helped us improving the paper. According to the main explanations, we made substantial changes in the manuscript and answered all comments, as you can find in detail below.

Thank you once again for the careful review process that improved a lot our manuscript.

Best regards,

Tania Araujo Jorge and Roberto Ferreira.

Comments and Suggestions for Authors:

The Authors have partially addressed key concerns but a few crucial corrections are needed (e.g. molecular function of deiodinases).

Response: Thanks for the comment. We provided a more specific description.

Also, it is still not included a polymorphism table that two Reviewers seem to agree as necessary - maybe Authors can request a deadline extension so the Table can be prepared accordingly.

 Response: Thanks for the comments. We included the table.

In addition, it is puzzling why the selenoprotein nomenclature remains inconsistent, when they cite the 2016 paper in which the standardization of selenoprotein names was determined. Consistent use of selenoprotein names should be achieved, and removal of old names, such as SEP15 (but not restricted to this one). Table 1, for example, has old selenoprotein names instead of the standardized ones.

Response: Thanks for the comment. We corrected it.

Comments on the Quality of English Language

The English improved from first version but still a few run-on sentences.

Response: Thanks for the comment. We hired a professional English proofreader, who made substantial changes to the text.

Reviewer 2 Report

Comments and Suggestions for Authors

i have no more comments

Author Response

Dear Reviewer,

We greatly appreciate your help in revising this manuscript.

Best regards,

Tania Araujo Jorge and Roberto Ferreira.